# Factors associated with adolescent pregnancy among Chepang women and their health-seeking behavior in Ichchhakamana rural municipality of Chitwan district

**Smriti Pant**[1]*, **Saugat Koirala**[2], **Anand Prasad Acharya**[3], **Pranil Man Singh Pradhan**[1]

1 Department of Community Medicine, Maharajgunj Medical Campus, Kathmandu, Nepal, 2 Department of Gynecology and Obstetrics, KIST Medical College, Imadol, Lalitpur, Nepal, 3 Department of Medicine, Frimley park Hospital, Frimley, Camberley, United Kingdom

* drsmritipant@gmail.com

## Abstract

Adolescent pregnancy is a critical public health issue, particularly in developing regions like Nepal, where it poses significant risks to maternal and child health and perpetuates the cycle of poverty. This study focused on the marginalized Chepang community, which is endangered and faces unique challenges. The study aimed to explore the factors associated with adolescent pregnancy among Chepang women in Ichchhakamana Rural Municipality, Chitwan, Nepal, and also assessed their reproductive health-seeking behavior. A cross-sectional analytical study was conducted with 217 Chepang women aged 15–20 years, and data was collected through face-to-face interviews using a semi-structured questionnaire. The collected data was entered and analyzed using IBM SPSS version 20. Descriptive statistical tools like frequency, and percentage were used to express the results. Pearson chi-square test, Fisher exact test were used for bivariate analysis to determine the presence of association between the dependent and independent variables. Binary logistic regression was used for further analysis. The prevalence of current adolescent pregnancy was 8.3%(18), while one-fourth had experienced prior pregnancies during their adolescence. Factors significantly associated with adolescent pregnancy included lack of education among the women and their mothers, as well as living in joint families. Additionally, number of antenatal visits and consumption of iron tablets seemed to be lower among Chepang women in comparison to the national data. Chepang women had high adolescent pregnancy rates, with low education level and joint family structure being important risk factors for it. They also had inadequate reproductive health seeking behavior. Addressing these problems requires strategies that prioritize education and raise awareness about reproductive health.

## Introduction

Adolescence is a state of transition for 10 to 19 year olds, during which many physical and psychological developments takes place [1]. When an adolescent gets pregnant it increases the

**Data Availability Statement:** All relevant data are within the manuscript and its Supporting information files.

**Funding:** The author(s) received no specific funding for this work.

**Competing interests:** The authors have declared that no competing interests exist.

risk of maternal and child mortality [2]. It is estimated that about 21 million adolescents get pregnant each year in developing regions, among which about 57% give birth [2]. Babies born to these adolescent mothers account for nearly 11% of births worldwide with 95% of them occurring in developing countries like Nepal [3]. Two million of these births are from girls under 15 years of age [4]. Complications related to pregnancy and childbirth are the leading cause of global deaths among adolescents of reproductive age group (15–19 years) [5]. Additionally, ninety-nine percent of the maternal deaths among 15 to 49-year-old women occur in low and middle income countries [5]. Furthermore, adolescent mothers aged 10–19 years face greater possibilities of occurrence of eclampsia, puerperal endometritis and systemic infections than women aged 20–24 years [6]. Other consequences are unsafe abortion, still birth and new born deaths [2].

There are many factors and determinants associated with adolescent pregnancy, explained in many literatures [3, 7, 8]. In case of marginalized community like Chepang community, factors associated with adolescent pregnancy are household wealth index, ethnicity, religion, sex of the household head, maternal education, literacy level and occupation [8]. Chepang community is a marginalized and endangered community of Nepal. According to National Population and Housing Census 2021, there are 84,364 Chepang inhabitants in Nepal out of which 27,643 live in Ichchhakamana Rural Municipality of Chitwan district [9]. The same report showed that 15.2% of the married 15 to 19-year-old Chepang women had at least one live birth in the last year [9]. Similarly, an article about adolescent pregnancy in Chepang women in 2016 illustrated that about 58.1 percent of the participants have their first child birth in their adolescent age [10]. It also showed that complications of adolescent pregnancy among them were preterm labour, prolonged labour, miscarriage and postpartum haemorrhage [10].

According to Nepal Demographic and Health Survey (NDHS) 2022, 14% of the women aged 15–19 years were ever pregnant [11]. This survey also shows that these pregnancies occurred more commonly among Muslims (22%) and Dalits (21%) [11]. Similarly, it illustrated that prevalence of pregnancy among 15 to 19 year olds was higher amid the less educated and amongst those who fall in the second wealth quintile [11]. NDHS 2022 estimates also show that both median age at first marriage and the median age of first sexual intercourse for women was 18.3 years [11]. These are risk factors for adolescent pregnancy. A study in Korak VDC in Chitwan showed that the average age of marriage in Chepang females was 15.48 years and mean age of first child birth was 16.95 years [10]. Similarly, another study in Dhading district of Nepal illustrated that the mean age of marriage and first pregnancy was 16.7 years and 17.7 years respectively [12]. Additionally, married Chepang women also had low contraceptive prevalence rate (49.9%) [12].

Health seeking behavior during pregnancy is very important as it determines the fate of not only the mother but also the child. Antenatal care (ANC), mode of delivery and postnatal care (PNC) are important aspects of health care that is necessary for pregnant women. One of the important components of the ANC recommendations is the number of ANC visits [13]. Even though the ANC model with proposed eight contacts between the healthcare provider and pregnant women, is considered superior in terms of pregnancy outcomes, many countries still follow the focused antenatal care (FANC) model which recommends 4 ANC visits during pregnancy [13]. Studies have shown that pregnant adolescents have more difficulty in accessing health services during pregnancy [14, 15]. A study conducted in Wakiso district in Uganda showed that pregnant adolescents were less likely to complete the recommended number of ANC visits when compared to their adult counterparts [14]. Similarly another study of adolescent mothers from Nigeria showed that difficulty in obtaining permission to visit the health service provider as well as far distance of the health center from home were two important factors that discouraged utilization of antenatal care [15].

There are many strategies adopted globally and by Nepal to reduce adolescent pregnancy. A global strategy for women's and children's health launched by United Nations Secretary General in 2010, stresses the importance of addressing the health and welfare of adolescent girls in order to achieve the goal on maternal mortality reduction [3]. Government of Nepal endorsed the 'National Adolescent Health and Development (NAHD) Strategy' in 2000 and developed an 'Implementation guideline on Adolescent Sexual and Reproductive Health (ASRH)' in 2007. Then in 2011, a National ASRH Program was designed and it has been slowly scaled up [16]. Nepal adopted the 'National Adolescent Development and Health Strategy 2075' in 2019, with the aim of fostering an environment where every Nepalese adolescent can lead a healthy, and productive life by embracing a positive lifestyle by the year 2025 AD [17].

This study aimed to determine the prevalence of adolescence pregnancy among Chepang women of Ichchhakamana Rural Municipality, Chitwan and to explore the factors associated with it. It also assessed their reproductive health seeking behavior.

## Materials and methods

### Study design and population

A cross-sectional analytical study was conducted in three wards of Ichchhakamana Rural Municipality, Chitwan. The data collection was done between July 2021 to October 2021. The study participants were Chepang women aged 15–20 years living at the study site.

The inclusion criteria included all Chepang women aged 15–20 years who gave consent to participate in the study while the exclusion criteria included, those who were unable to answer due to language barrier. The sample size for the study is calculated by using the formula,

$$\text{Sample size } (n) = z^2 pq/d^2$$

where, z = standard normal variable at 95% confidence level (1.96)

p = expected proportion in population based on prior studies = 0.17 i.e., 17% prevalence of teenage pregnancy [18]

q = 1-p = 1–0.17 = 0.83
d = 0.05 (5% margin of error)
hence,

$$n = \left\{ (1.96)^2 \, X \, 0.17 \, X \, 0.83 \right\} / (0.05)^2$$
$$= 216.8$$

Considering 10% non- response rate, total sample size = 216.8 / (0.9) = 240.8≈241
Therefore, the total sample size was 241.

The study started with purposive selection of Ichchhakamana Rural Municipality as the study site. The municipality consists of seven wards. At the first stage, three wards were chosen randomly for the study using lottery method. Next, the study site was visited and the proposed participants in the Chepang community were informed about the objectives of the research and its expected outcome. After getting approval for engagement, the required sample was conveniently taken from the selected three wards (80 samples each from wards 1 and 3, and 81 samples from ward 6). First house from respective ward was selected randomly and rest of the houses were selected consecutively until required sample size was reached. If more than one, participant meeting the eligibility criteria were found at selected households then all of them were interviewed.

## Study variables and data collection

The dependent variable was adolescent pregnancy while independent variables included socio-demographic, sexual and reproductive characteristics of the participants and reproductive health seeking behavior (S1 Table).

Adolescent pregnancy was defined as any woman aged 15–20 years who had given birth within the last 1 year or any woman aged 15–19 years who was pregnant during the time of the study.

Reproductive health seeing behavior was defined as any action or inaction taken by pregnant adolescents for pregnancy related issues like confirming diagnosis of pregnancy, making antenatal (ANC) visits, consumption of iron and calcium tablets and undergoing various assessments (like physical examination, blood tests, urine tests and ultrasonography).

The data was collected with the help of a semi-structured questionnaire by using face to face interview technique. The questionnaire consisted of information regarding socio-demographic details, sexual and reproductive characteristics, and reproductive health seeking behavior. Related literature was reviewed for making the questionnaire [10, 12, 14, 15]. The questionnaire was shared with content experts for content validity. Face validity of the questionnaire was maintained by pretesting the questionnaire (among the population in the selected ward who do not belong to Chepang ethnicity). After pretesting, necessary modifications were made to the questions. As the questionnaire demands recalling information for the past 1 year, there is a chance of recall bias.

## Statistical analysis

The collected data was entered and analyzed using IBM SPSS version 20. First of all, the prevalence of adolescent pregnancy was calculated. Descriptive statistical tools like frequency, and percentage were used to express the results. Pearson chi-square test, Fisher exact test were used for bivariate analysis to determine the presence of association between the dependent and independent variables. Crude odds ratio (COR) at 95% confidence interval (95% CI) was calculated to see the magnitude of association of adolescent pregnancy with independent variables. All the independent variables that were associated with adolescent pregnancy status with a significance level of less than 0.20 (p-value< 0.20) were included in multivariable analysis. As the dependent variable was dichotomous and independent variables were categorical, binary logistic regression was carried out to calculate corresponding adjusted odds ratio (AOR). All tests were done with the significance level set at 5% (p-value <0.05). In the assessment of multi-collinearity, the Variance Inflation Factor (VIF) was determined to be less than 10 (S2 Table). This confirms the absence of any interconnection among the independent variables.

## Ethical consideration

Ethical approval was taken from Nepal Health Research Council (ERB Registration Number is 329/2021 P). The office of Ichchhakamana rural municipality was contacted and a letter of support for community engagement was obtained. After this, the ward chairpersons of the selected wards were contacted and permission was taken to conduct the research. Next, a written informed consent was obtained from the participants and/or their parents. If the participants were either married or of legal age at the time of interview, informed consent was obtained from the participants themselves. However, if the participants were either unmarried or minors, informed consent was obtained from the parent and assent was taken from the participant. The participants' participation in the study was voluntary and their confidentiality

was maintained. The information will be kept safely in a computer with proper password to disallow any fallacy in conduction of the study.

### Inclusivity in global research

Additional information regarding the ethical, cultural, and scientific considerations specific to inclusivity in global research is included in the Supporting Information (S1 Checklist).

## Results

Out of the 241 women selected, a total of 217 participants responded, yielding a response rate of 90%. The features of the participants are as follows:

### Socio-demographic characteristics of the participants

They were aged between 15 to 20 years of age. Their mean age was 17.6 years with standard deviation of 1.7 years. Most of the participants were aged 18 years (19.8%) followed by 20 years of age (Table 1). Furthermore, most of the participants were Hindu (85.3%) and majority of the women had studied till primary level (67.7%) (Table 1). Similarly, 30% were involved in farming (Table 1).

More than half (54.8%) of the families were nuclear (Table 1). In about three quarters (73.3%) of the families, the head of the household was a male. Additionally, 64% of the families had an income of more than NRs. 15,000 (Table 1). Similarly, almost half of the participant's mother had never been to school while the rest had studied till primary level (Table 1).

### Sexual and reproductive health characteristics of the participants

Almost half (47.5%) of the participants were married and majority (28%) of them were married at 18 years of age (Table 2). The mean age of marriage was 16.84 years with standard deviation of 1.52 years. About 48% of the participants stated that they have been sexually active (Table 2). For majority of them (64.4%), the age of first sexual contact was 15 to 18 years and only about 14% had used contraceptive devices (Table 2). The most common reason behind not using contraceptive methods were its use not being feasible (42.7%), desire to get pregnant (36%) and lack of awareness (15.7%) about the methods (Table 2).

### Adolescent pregnancy status

Among the 217 participants, 8.3% (18) were currently pregnant or had delivered a baby in the last one year. Almost all except one pregnancy was unplanned. But all of the pregnant women were happy about their pregnancy. Amongst the pregnant, most (44.5%) were in their second trimester. Almost a quarter (24%) of the participants had a previous pregnancy. Among them, 79% had live births and majority of them did not have any health issues among mothers (90.2%) or the neonates (82.7%) at the end of the pregnancy.

### Reproductive health seeking behavior

Among those who were currently pregnant, 72% women considered amenorrhea as an indicator of pregnancy, while the rest relied on health checkup for the diagnosis of pregnancy (Table 3). Additionally, half of the pregnant women had only one antenatal (ANC) visit, while none of them had done four visits (Table 3). Furthermore, 50% of the pregnant women were taking iron tablets, but only 22% were taking calcium tablets (Table 3). Regarding the tests done during pregnancy, 83.3% of the women had physical examination,66.7% had a urine examination, 22% had an ultrasonography, 11% had done blood tests, but only 11% of them had done all of these.

## Association of adolescent pregnancy with socio-demographic variables

Higher percentage of participants who were pregnant belonged to Hindu religion. But this relationship was not statistically significant (Table 4). The study showed that those adolescents who had never been to school had 2.9 times more chance of being pregnant compared to those

**Table 1. Socio-demographic characteristics of the participants.**

| Variables | Frequency | Percentage |
|---|---:|---:|
| **Age (in years)** | | |
| 15 | 35 | 16.1 |
| 16 | 41 | 19.0 |
| 17 | 22 | 10.0 |
| 18 | 43 | 19.8 |
| 19 | 35 | 16.1 |
| 20 | 41 | 19.0 |
| Total | 217 | 100.0 |
| **Religion** | | |
| Hindu | 185 | 85.3 |
| Christian | 32 | 14.7 |
| Total | 217 | 100.0 |
| **Education** | | |
| Never been to school | 59 | 27.0 |
| Primary Level | 147 | 68.0 |
| SEE pass | 7 | 3.2 |
| Intermediate level | 4 | 1.8 |
| Total | 217 | 100.0 |
| **Occupation** | | |
| Housewife | 45 | 20.7 |
| Farmer | 70 | 32.3 |
| Government Job | 1 | 0.4 |
| Private Job | 3 | 1.4 |
| Laborer | 54 | 24.9 |
| Student | 44 | 20.3 |
| Total | 217 | 100.0 |
| **Family Type** | | |
| Nuclear | 119 | 54.8 |
| Joint Family | 98 | 45.2 |
| Total | 217 | 100.0 |
| **Head of the Family** | | |
| Male | 159 | 73.3 |
| Female | 58 | 26.7 |
| Total | 217 | 100.0 |
| **Mother's educational status** | | |
| Never been to school | 111 | 51.2 |
| Primary Level and above | 106 | 48.8 |
| Total | 217 | 100.0 |
| **Family Income (Monthly)** | | |
| Less than Rs. 15000 | 78 | 35.9 |
| More than or equal to Rs. 15000 | 139 | 64.1 |
| Total | 217 | 100.0 |

**Table 2. Sexual and reproductive health characteristics.**

| Variables | Frequency | Percentage |
|---|---:|---:|
| **Marital status** | | |
| Single | 113 | 52.0 |
| Married | 103 | 47.5 |
| Widower | 1 | 0.5 |
| Total | 217 | 100.0 |
| **Age at Marriage (in years)** | | |
| 14 | 4 | 3.8 |
| 15 | 25 | 24.0 |
| 16 | 27 | 26.0 |
| 17 | 10 | 9.6 |
| 18 | 29 | 28.0 |
| 19 | 4 | 3.8 |
| 20 | 5 | 4.8 |
| Total | 104 | 100.0 |
| **Sexual relationship status** | | |
| Had sexual relations | 104 | 47.9 |
| Never had sexual relations | 113 | 52.1 |
| Total | 217 | 100.0 |
| **Age at first sexual relationship(in years)** | | |
| Less than15 | 25 | 24.0 |
| 15 to 18 | 67 | 64.4 |
| More than 18 | 12 | 11.6 |
| Total | 104 | 100.0 |
| **Use of contraceptive** | | |
| Ever used contraceptive | 15 | 14.4 |
| Never used contraceptive | 89 | 85.6 |
| Total | 104 | 100.0 |
| **Reasons for not using contraceptives** | | |
| Not feasible | 38 | 42.7 |
| Desire to get pregnant | 32 | 36.0 |
| Lack of awareness | 14 | 15.7 |
| Living separately from spouse | 3 | 3.4 |
| Family influence | 2 | 2.2 |
| Total | 89 | 100.0 |

who had been to at least primary school (COR (95%CI): 2.9[1.1–7.9]) and this relationship was statistically significant. (p <0.05) (Table 4). Similarly, those who lived in joint families had a higher chance of being pregnant compared to nuclear families (COR (95%CI): 4.8 [1.5–15.0]) and this relationship was also statistically significant (p<0.05) (Table 4). Among the pregnant adolescents most had females as head of the family and lived in families who earned less than NRs. 15,000. But these associations were not statistically significant. Furthermore, adolescents with mothers who never went to school had 3.6 times more risk of being pregnant compared to those who had primary level education (COR, 95% CI: 3.68[1.17–11.57]) and this association was statistically significant (p<0.5).

After adjustment for confounders only living in joint family (AOR, 95% CI: 3.4[1.01–11.74]) was significantly associated with higher occurrence of adolescent pregnancy.(p<0.05)

**Table 3. Reproductive health seeking behavior of pregnant adolescents.**

| Variables | Frequency | Percentage |
|---|---:|---:|
| **Method of diagnosis of pregnancy** | | |
| Amenorrhea | 13 | 72.0 |
| Check up at health facility | 5 | 28.0 |
| Total | 18 | 100.0 |
| **ANC Visits during pregnancy** | | |
| Once | 9 | 50.0 |
| Twice | 3 | 16.7 |
| Thrice | 4 | 22.2 |
| Never | 2 | 11.1 |
| Total | 18 | 100.0 |
| **Consumption of iron during pregnancy** | | |
| Yes | 9 | 50.0 |
| No | 9 | 50.0 |
| Total | 18 | 100.0 |
| **Consumption of calcium during pregnancy** | | |
| Yes | 4 | 22.2 |
| No | 14 | 77.8 |
| Total | 18 | 100.0 |

## Discussion

### Socio-demographic characteristics

Among the 217 participants, majority of the women had studied till primary level (67.7%) while the rest had no education. This is in contrast to a study among 80 Chepang women in Dhading district of Nepal, where almost 26 percent had studied till primary level, but almost 30 percent had education of secondary level or higher [12]. Similarly another study in Korak VDC, Nepal among 148 Chepang women showed that 36.4% of the women had primary level education and almost 5% had secondary level education [10]. Furthermore, the most common occupation among the participants in our study was farming, which was similar to the study in Korak VDC but in contrast to the finding in Dhading where most of the participants were housewives [12]. In our study more than half of the families were nuclear. This finding was different than other studies conducted in Nepal [10, 12]. Additionally, in this study, most of the families had an income of more than NRs. 15,000. This was also in contrast to the findings of another study in Nepal [12].

In our study, the mean age of marriage was 16.84 years and almost half of the participants stated that they were sexually active. For majority of them (64.4%), the age of first sexual contact was 15 to 18 years. This is in contrast to the findings of NDHS 2022, according to which both the age at marriage and age at first sexual contact was 18.3 years for females [11].

### Adolescent pregnancy

In our study, the prevalence of adolescents who were currently pregnant was 8.3% (18). This is in contrast to the results of a systematic review and meta-analysis conducted in Nepal which showed the prevalence to be a bit higher (13.2%) [19]. Similarly, a report by UNICEF showed that, in South Asia region adolescent marriage and pregnancy was highest among Nepalese and Bangladeshi girls [20]. Furthermore, a noteworthy finding from our study is that approximately one-fourth of the participants disclosed a history of prior pregnancies. Given the age

Table 4. Association of adolescent pregnancy with socio-demographic variables.

| Variables | Pregnant | Not Pregnant | Total | COR[95%CI] | p value | AOR[95%CI] | p value |
|---|---|---|---|---|---|---|---|
| **Religion** | | | | | | | |
| | | | | | | | |
| Hindu | 16(8.6) | 169(91.4) | 185(100.0) | 1.4 [0.3–6.4] | 1.0[a] | - | - |
| Christian | 2(6.3) | 30(93.8) | 32(100.0) | Ref | | - | - |
| **Participant's Education level** | | | | | | | |
| Never been to school | 9(15.3) | 50(84.7) | 59(100.0) | 2.9 [1.1–7.9] | 0.048[a]* | 1.8[0.61–5.5] | 0.28 |
| Primary level and above | 9(5.7) | 149(94.3) | 158(100.0) | Ref | | | |
| **Family Type** | | | | | | | |
| Joint | 14(14.3) | 84(85.7) | 98 (100.0) | 4.8 [1.5–15.0] | 0.004[b]* | 3.4[1.01–11.74] | 0.049* |
| Nuclear | 4(3.4) | 115(96.6) | 119(100.0) | Ref | | Ref | |
| **Head of the Family** | | | | | | | |
| Female | 8(13.8) | 50(86.2) | 58(100.0) | 2.38[0.89–6.37] | 0.095[a] | 0.91[0.28–2.8] | 0.86 |
| Male | 10(6.3) | 149 (93.7) | 159(100.0) | Ref | | Ref | |
| **Monthly Family Income** | | | | | | | |
| < NRs. 15,000 | 8(10.3) | 70(89.7) | 78 (100.0) | 1.47[0.56–3.91] | 0.43[b] | - | - |
| ≥ NRs. 15,000 | 10(7.2) | 129(92.8) | 139 (100.0) | Ref | | - | - |
| **Mother's Education level** | | | | | | | |
| Never been to school | 14(12.6) | 97(84.4) | 111(100.0) | 3.68[1.17–11.6] | 0.018[b]* | 2.23[0.64–8.18] | 0.2 |
| Primary level and above | 4(3.8) | 102(96.2) | 106(100.0) | Ref | | Ref | |

[a] Fisher exact test

[b] Chi-square test

*statistically significant at p<0.05

range of our participants (15 to 20 years), this observation suggests that even the women who were not pregnant during the study duration experienced teenage pregnancies at some stage during their adolescence.

## Reproductive health seeking behavior

In our study, half of the pregnant women had only one ANC visit, while none of them had gone for the recommended four ANC visits. This is in contrast to our national data of fiscal year 2078/79, which estimates that 79.4% of the women in reproductive age had four ANC visits [21]. Furthermore, 50% of the pregnant women were taking iron tablets, but only 22% were taking calcium tablets. This is in contrast to the NDHS 2022 findings, according to which 96% of the pregnant women took iron tablets or syrup in their last pregnancy [11]. These variations may be due to the fact that Chepang is a marginalized community and has less access to health services.

## Association of adolescent pregnancy with socio-demographic variables

Our study showed that those adolescents who had never been to school had 2.9 times more chance of being pregnant compared to those who had been to at least primary school (COR (95%CI): 2.9[1.1–7.9]). This finding is supported by World Bank estimates which states that more education for girls can result in delayed pregnancy [22]. Similarly, several other research has illustrated that marrying at a young age and early pregnancy were both results of low level of education among the adolescent girls [20, 23, 24]. Education empowers women by providing them with knowledge about reproductive health and family planning, leading to more

informed choices [25]. This may result in delayed marriages, encourage aspirations beyond early motherhood, and enhance access to resources and healthcare.

Furthermore, in our study those who lived in joint families had a higher chance of being pregnant compared to nuclear families (COR (95%CI): 4.8 [1.5–15.0]) and this relationship was also statistically significant (p<0.05) (Table 4). The higher likelihood of pregnancy in joint families can be attributed to different reasons. Residing in joint families can result in limited personal freedom and the perpetuation of traditional beliefs, often leading to early age at marriage. Additionally, if the joint family is from a low resource setting, it will have limited space and resources, which motivates the family to marry off their daughters at a very early age [26]. This ultimately leads to early childbearing among adolescents. On the contrary, joint family also offers the advantage of a larger support system and sharing of responsibilities [27], which may promote pregnancies by creating a more conducive environment for giving birth to and raising children.

Adolescents with mothers who never went to school had 3.6 times more risk of being pregnant compared to those who had at least primary level education (COR, 95% CI: 3.68[1.17–11.57]) and this association was statistically significant (p<0.5). Mothers' education has been associated with reduced level of fertility and increased age at first birth for her offspring [28]. Hence, this heightened risk of pregnancy among the adolescents may be attributed to limited knowledge of reproductive health among the uneducated mothers.

The strength of our study lies in its community based approach, which greatly increases the likelihood of capturing the real scenario of adolescent pregnancy among the Chepang women.

Our study has some limitations. Firstly, as the participants had to recollect memories of the last one year in order to respond to the questions, recall bias may have occurred. Additionally, due to the limitation of funding only three wards were chosen from Ichchhakamana Rural Municipality. Lastly, as the sample size was small the results cannot be generalized to the whole country. Hence, a comprehensive national study, involving a representative sample, is necessary to delve deeply into the issue of adolescent pregnancy within the Chepang community.

## Conclusion

Adolescent pregnancy was high among the Chepang women in Ichchhakamana Rural Municipality. Low level of education among women and their mothers, along with their family type seemed to be important risk factors for adolescent pregnancy. Similarly, the Chepang women exhibited inadequate health seeking behavior. Government should develop and implement strategies that encourage education for girls especially in marginalized communities like the Chepang, so that they are empowered to make better choices for their lives. Similarly, recognizing the pivotal role family plays in this matter, raising community awareness regarding reproductive health issues may also help address the problem of adolescent pregnancy.

## Supporting information

**S1 Table. Variables of the study.**
(DOCX)

**S2 Table. Collinearity statistics.**
(DOCX)

**S1 Checklist. Inclusivity in global research.**
(DOCX)

**S2 Checklist. STROBE statement.**
(DOCX)

**S1 File. Master sheet.**
(XLSX)

## Acknowledgments

We wish to extend our profound appreciation to all adolescents and their families for their participation and invaluable support. Additionally, we would like to recognize the assistance rendered by the ward representatives of wards one, three, and six within the Ichchhakamana Rural Municipality. Their authorization for data collection and cooperation are highly acknowledged and appreciated.

## Author Contributions

**Conceptualization:** Smriti Pant, Saugat Koirala, Anand Prasad Acharya, Pranil Man Singh Pradhan.

**Data curation:** Smriti Pant.

**Formal analysis:** Smriti Pant.

**Investigation:** Smriti Pant, Saugat Koirala, Anand Prasad Acharya.

**Methodology:** Smriti Pant, Saugat Koirala, Anand Prasad Acharya, Pranil Man Singh Pradhan.

**Resources:** Smriti Pant.

**Supervision:** Smriti Pant, Pranil Man Singh Pradhan.

**Validation:** Smriti Pant, Pranil Man Singh Pradhan.

**Writing – original draft:** Smriti Pant.

**Writing – review & editing:** Smriti Pant, Saugat Koirala, Pranil Man Singh Pradhan.

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
