## [Decision Letter · Decision Letter 0]

30 Oct 2023

PONE-D-23-26050Factors Associated with Adolescent Pregnancy among Chepang Women and Their Health-Seeking Behavior in Ichchhakamana Rural Municipality of Chitwan DistrictPLOS ONE

Dear Dr. Pant,

Thank you for submitting your manuscript to PLOS ONE. After careful consideration, we feel that it has merit but does not fully meet PLOS ONE’s publication criteria as it currently stands. Therefore, we invite you to submit a revised version of the manuscript that addresses the points raised during the review process.

We look forward to receiving your revised manuscript.

Kind regards,

Umesh Raj Aryal, PhD

Academic Editor

PLOS ONE

Journal Requirements:

3. Thank you for submitting the above manuscript to PLOS ONE. During our internal evaluation of the manuscript, we found significant text overlap between your submission and previous work in the [introduction, conclusion, etc.].

Please revise the manuscript to rephrase the duplicated text, cite your sources, and provide details as to how the current manuscript advances on previous work. Please note that further consideration is dependent on the submission of a manuscript that addresses these concerns about the overlap in text with published work.

[If the overlap is with the authors’ own works: Moreover, upon submission, authors must confirm that the manuscript, or any related manuscript, is not currently under consideration or accepted elsewhere. If related work has been submitted to PLOS ONE or elsewhere, authors must include a copy with the submitted article. Reviewers will be asked to comment on the overlap between related submissions (http://journals.plos.org/plosone/s/submission-guidelines#loc-related-manuscripts).]

We will carefully review your manuscript upon resubmission and further consideration of the manuscript is dependent on the text overlap being addressed in full. Please ensure that your revision is thorough as failure to address the concerns to our satisfaction may result in your submission not being considered further.

6. Please include a copy of Table 7 which you refer to in your text on page 14.

7. We are unable to open your Supporting Information file [Chepang Women 15th Aug.sav]. Please kindly revise as necessary and re-upload.

Additional Editor Comments:

Method section

The sample size calculation is not clear. Please clarify it. what about the non-response rate?

How was the sampling interval computed? Provide its details.

Clarify the adolescent pregnant definition. 15-19 and 15-20 years make confusion.

In Line 120, it should provide detailed information but not "etc".

The title of Table 1 does not match as the table is broken down into different subheadings.

Please revise it. On what basis is income divided by 15000?

"Never had a sexual relationship" how is it linked with adolescent pregnancy? it also impacts your analysis.

"extensive literature review was done for making the questionnaire". Provide reference.

How did you ensure face validity?

What did the authors do to reduce recall bias since they knew it in advance ?

Introduction

How does it lead to a vicious cycle of ill health and poverty?

Discuss a few words about the Chepang community before jumping into the factors.

Is there a reason to focus on NDHS 2016 rather than 2020?

Line 68-69 move to the method section and link with sample size calculation.

Nothing is mentioned about the health-seeking behavior of adolescent pregnancy.

Analysis

Though the author has tried to explain details about the analysis but it is unclear to readers. Therefore authors are required to provide with rationale behind it as well as a step-by-step procedure. Do not get confused with multivariable and multivariate analysis. Provide details of VIF results.

Ethics

It is not clear with whom the researchers take the assent and consent form. It needs to be discussed in detail. The next definition of parents for unmarried and married should be clarified with details.

Results

There is no table that explains the details of the crude odds ratio as the authors directly jump into table 4. So the table for bi-variate analysis is to be presented.

Most of the results of table 3 are explained in text so it is suggested to remove table 3.

Discussion

Discuss the strengths of the study prior to the limitations.

Need to revised discussion section as repeation of result section.

Reviewers' comments:

Reviewer's Responses to Questions

**Comments to the Author**

1. Is the manuscript technically sound, and do the data support the conclusions?

Reviewer #1: Yes

Reviewer #2: Partly

Reviewer #3: Partly

2. Has the statistical analysis been performed appropriately and rigorously? 

Reviewer #1: Yes

Reviewer #2: No

Reviewer #3: Yes

3. Have the authors made all data underlying the findings in their manuscript fully available?

Reviewer #1: Yes

Reviewer #2: Yes

Reviewer #3: Yes

4. Is the manuscript presented in an intelligible fashion and written in standard English?

Reviewer #1: Yes

Reviewer #2: No

Reviewer #3: Yes

5. Review Comments to the Author

Reviewer #1: Introduction: The research topic is very relevant in the Nepalese context, the introduction part is well written on the basis of international, national, and local context with comparing data. The objective of the study has been included in the introduction.

Materials and Methods: The research design, study population, and sampling technique were well-explained. The sample size should be properly explained in simple manner.

Figures, tables, results:

In Table 1, there were too many rows, it could be broken into two tables (Socio-demographic and Sexual and Reproductive health characteristics).

In topic/title "Health Seeking Behaviour" is mentioned as the objective of the study but a table/figure about this is displayed.

Some sentences have been written repeatedly.

Author guidelines should be followed for writing the table's title and illustration.

Discussions: The discussions section is good, relation literature were compared and contrasted in the different scenario but some literature were repeated. The reproductive health-seeking behavior was also not discussed in the discussion section, the topic is given less priority in the study. It would be better to include the source of information regarding the RH.

Conclusions: The conclusion is well-written, try to elaborate a little.

Reviewer #2: Abstract:

• Suggest using health-seeking behavior, adolescent sexual and reproductive health, teenage pregnancy, influencing factors, and the Chepang populationas key words for the paper.

• There are some critical concerns to address. First, the age group chosen for the study, 15-20 years, seems to differ from commonly used definitions of adolescence (15-19 years), and this warrants clarification. The inclusion of women aged 20 or above in the study should be explained, or alternatively, the data analysis should be revised to maintain consistency with established age categories for adolescents.

• Additionally, while the abstract highlights the role of education as a significant factor in teenage pregnancy among the Chepang community, this point isn't adequately supported in the results and discussion section of the manuscript. It's essential to align the abstract with the study's findings and offer a brief outline of the methodology, including statistical significance.

Introduction:

• The introduction should provide a clearer and more coherent background context for the study. It currently relies on outdated data from NDHS 2016 when more recent data is available (NDHS 2021). If this study predates the NDHS 2021, considering the MICS 2019 results for the problem statement might be more appropriate. It's also suggested to offer a comparative overview of adolescent pregnancy in Nepal, supported by evidence and factors associated with it. Please refer this link https://www.prb.org/wpcontent/uploads/2022/02/Adolescent_Fertility_Atlas_Nepal.pdf for getting more overview of the adolescent fertility.

• Furthermore, for the Chepang population, it is essential to verify information with the National Population and Housing Census 2021 report. The study title emphasizes health-seeking behavior, but this connection to adolescent pregnancy and relevant influencing factors needs to be more explicitly established.

Materials and Methods:

• The choice of the 15-20 age group for the study should be rationalized, and bias control strategies for age group should be explained. The significance level of p=0.17 requires clarification. The purpose of selection of the study site and the choice of only 3 wards from 7 wards should be justified.

• To enhance reader-friendliness, provide an overview of variables in table form with operational definitions for both dependent and independent variables. Clarify the criteria for defining adolescent pregnancy as “adolescent age of 15-20”, ensuring technical accuracy.

• The sample population details should specify whether only married adolescent girls were included or if unmarried individuals were considered and the reasons for this choice.

Results:

• Clarify whether the contraceptive use percentages are among married adolescent girls/women. Provide the number of adolescent pregnant individuals, which is currently absent in the table. Explain the ANC status among pregnant adolescents and relate it to national protocols on pregnancy and antenatal care indicators. This will help to relate health-seeking behaviour among the adolescent for their care and wellbeing.

• When discussing the association of adolescent pregnancy with socio-demographic and family variables, acknowledge the wider confidence intervals and the limitations caused by the small sample size, if relevant. Specify if the analysis is based solely on married participants.

• The examination of health-seeking behavior is currently missing, this should be more detailed.

Discussion:

• The discussion should offer a more in-depth analysis of the factors influencing teenage pregnancy, connecting them to previous studies. Address statistically insignificant relationships between variables and be honest to explain data, methodology limitations and recommend the need for further research. The sample size is too small to generalize the result.

• • Teenage pregnancy is often associate with the child marriage in the context like Nepal and child marriage is a result of lack of access to education, information and rights, sexual and reproductive health services etc. which influences individual behaviours. It is important to discuss why child marriage is prevalent there. How has education affected adolescent’s and their family’s health-seeking behaviour? What makes them happy to have a baby as their early age? As health-seeking behaviour is related to individual and community socio-economic and cultural factors. Also, the husband’s education, economic status affects the adolescent pregnancy and health-seeking behaviour which have not been assessed while mother's education and its association with their daughter teenage pregnancy has been briefly touched. I suggest considering all these factors to make a strong argument and scientific explanation for the recommendation. I suggest the author visit global and national report to comprehend the knowledge with the research findings. Include possible alternative explanations and elaborate on policy recommendations, as initially intended.

• Include possible alternative explanations and elaborate on policy recommendations, as initially intended.

References:

• Ensure that the citation and referencing align with the journal's requirements, preferably using the Vancouver style. Rely on more scientific articles than newspaper to strengthen the arguments and consider thorough English language editing for better comprehension and logical flow.

Reviewer #3: Dear authors,

Congratulations for the efforts of writing such an interesting topic regarding the adolescent women of Chepang community.

Some of the portion of research methodology need to be clearly explained (inclusion criteria specially) and the result section.

Suggestions have been made in the word file of the manuscript, please view it.

6. PLOS authors have the option to publish the peer review history of their article (what does this mean?). If published, this will include your full peer review and any attached files.

Reviewer #1: No

Reviewer #2: **Yes: **Adweeti Nepal

Reviewer #3: **Yes: **Chandra Kumari Garbuja

---

## [Author Response · Author response to Decision Letter 0]

31 Dec 2023

Reply to Editor’s Comments

All the journal requirements have been fulfilled:

1. The article has been revised to fulfil the journal requirements.

2. A copy of PLOS’ questionnaire on inclusivity in global research has been added to the online submission system.

3. Regarding the overlap of the text. We understand the concept of plagiarism and how important it is to cite scientific work. We have cited all our sources. We have also tried to paraphrase and not copy the information from our sources. 

4. Orchid ID of the corresponding author has been updated

5. The information regarding ethical approval is in Lines 164-173. There also seemed to be confusion regarding the type of consent taken form the participants. This part has been more elaborately explained.

6. Table 7 was quoted by error. This was earlier merged into another table. 

7. As the SPSS file titled ‘Chepang women.sav’ could not be opened, an excel file (Master sheet Chepang Women) with the same information has been uploaded. 

Additional Editor Comments

Method section

Q1: The sample size calculation is not clear. Please clarify it. what about the non-response rate?

A1: The calculation steps have been clarified.

How was the sampling interval computed? Provide its details.

The sample size for the study is calculated by using the formula,

Sample size (n) = z2pq / d2

where,

z = standard normal variable at 95% confidence level (1.96)

p = expected proportion in population based on prior studies = 0.17 i.e., 17% prevalence of teenage pregnancy .

q = 1-p = 1-0.17 = 0.83

d = 0.05 (5% margin of error)

hence,

n = {(1.96)2 x 0.17 x 0.83}/(0.05)2

= 216.82= 217

 This has been included in the paper.

Q2: Clarify the adolescent pregnant definition. 15-19 and 15-20 years make confusion.

A2: Adolescent pregnancy percentage is commonly defined as “Percentage of women aged 15-19 who have given birth or are pregnant with their first child”. As our study measures the prevalence of adolescent pregnancy in the last one year, the operational definition of adolescent pregnancy for this study is as follows:

a. Women aged 15-20 years who have had a live birth within the last 1 year OR

 b. Women aged 15-19 years who are pregnant with their first child during the time of the study 

 The reason our study considers women up to 20 years of age is as follows:

As we are measuring current age of the participants, women who were 19 years of age and delivered last year may be 20 years of age at the time of the study. To include those women, the population group has been mentioned as women aged 15 to 20 years of age. But in essence, the study will still include only adolescents as those women were 19 years old when they delivered.

Q3: In Line 120, it should provide detailed information but not "etc".

A3: The correction has been made. 

Q4: The title of Table 1 does not match as the table is broken down into different subheadings.

Please revise it. 

A4: The table has been broken down and the table title revised.

Q5: On what basis is income divided by 15000?

A5: During pretesting of the questionnaire the average income was found to be Rs. 15000.

Q6: "Never had a sexual relationship" how is it linked with adolescent pregnancy? it also impacts your analysis.

A6: This question was kept in order to address the sexual and reproductive characteristics of the study population, which is one of the objectives of the study.

Q7: "extensive literature review was done for making the questionnaire". Provide reference.

A7: References have been added.

Q8: How did you ensure face validity?

A8: For face validity of the questionnaire pretesting was done in 10 % of sample size (among the population in the selected ward who do not belong to Chepang ethnicity) and necessary modifications were made to the questions.

Q9: What did the authors do to reduce recall bias since they knew it in advance?

A: The authors made sure that the participants had enough time to recall their memories and also probed them by trying to link the event with the timeline of some other event that may have occurred during the same time.

Introduction

Q10. How does it lead to a vicious cycle of ill health and poverty?

A10: This portion has been removed while revising the introduction section.

Q11. Discuss a few words about the Chepang community before jumping into the factors.

A11. The text in introduction has been re-arranged to include information about Chepang people earlier.

Q12. Is there a reason to focus on NDHS 2016 rather than 2020?

The writing of the article was started before the publication of the NDHS 2022 reports. However, while revising the Introduction, new data has been added from NDHS 2022.

Q13. Line 68-69 move to the method section and link with sample size calculation.

A13. This portion has been rearranged in the introduction part in response to Q 11.

Q14.Nothing is mentioned about the health-seeking behavior of adolescent pregnancy.

A14. This portion has been added. Lines 75-88.

Analysis

Q15. Though the author has tried to explain details about the analysis but it is unclear to readers. Therefore authors are required to provide with rationale behind it as well as a step-by-step procedure. Do not get confused with multivariable and multivariate analysis. Provide details of VIF results.

A15. Changes have been made. Lines 156-162.

Ethics

Q16. It is not clear with whom the researchers take the assent and consent form. It needs to be discussed in detail. The next definition of parents for unmarried and married should be clarified with details.

A16: This has been clarified in lines 164-173.

Results

Q17: There is no table that explains the details of the crude odds ratio as the authors directly jump into table 4. So the table for bi-variate analysis is to be presented.

A17: The Crude Odds ratio (COR) and its p value has been depicted in Table 4. Since there is a limitation to the number of tables that can be incorporated in the manuscript, the information of both COR and AOR has been incorporated into one single table.

Q18: Most of the results of table 3 are explained in text so it is suggested to remove table 3.

A18: The old table 3 has been removed and replaced by another table.

Discussion

Q19. Discuss the strengths of the study prior to the limitations.

A19: This has been done.

Q20. Need to revised discussion section as repetition of result section.

A20: This has been done.

Reply to comments from Reviewer 1

I. Reply to overall comments:

Materials and Methods: 

Q1: The sample size should be properly explained in simple manner.

A1: This has been explained using the formula for sample size calculation in lines 109-118.

Figures, tables, results:

Q2: In Table 1, there were too many rows, it could be broken into two tables (Socio-demographic and Sexual and Reproductive health characteristics).

A2: This has been done. The table has been broken down to two parts.

Q3: In topic/title "Health Seeking Behaviour" is mentioned as the objective of the study but a table/figure about this is displayed.

A3: The information is displayed in table 3 and a separate paragraph has been added for its description. 

Q4: Some sentences have been written repeatedly.

A4: The whole text of the manuscript has been read thoroughly and revised to avoid duplication.

Q5: Author guidelines should be followed for writing the table's title and illustration.

A5: Necessary changes have been made.

Discussions: 

Q6: The reproductive health-seeking behavior was also not discussed in the discussion section, the topic is given less priority in the study. It would be better to include the source of information regarding the RH.

A6: Information about Health Seeking behavior has been added. Lines 275-281.

Q7: Conclusions: The conclusion is well-written, try to elaborate a little.

A7: Conclusion section has been revised.

II. Reply to comments in word file

Introduction:

Q8. It would be better to write less study…..

A8: This line has been removed while revising the introduction part.

Q9: It seems related article

A9: Yes. It has been included as it includes information about birth among adolescents in the Chepang community.

Q10: This study is less appropriate here.

A10: This portion has been removed.

Methodology

Q11. Write in proper sentence, it would give another sense too.

A11. The sentence has been revised.

Q12. This sentence has already discussed above.

A12. This has been mentioned as an operational definition for the study.

Q13: Better to mention type of questionnaire

A13: The questionnaire was a semi-structured in nature. This has been mentioned in the text. The details of the questionnaire is described in paragraph that follows. Lines 141-144.

Q14: I think this process is understood not necessary to write here

A14: This has been mentioned to highlight the ethical aspect of the study.

Q15. Already discussed

A15: This portion has been removed.

Results:

Q16: This table is too long, too many rows it can be broken into two tables. Pls follow the guidelines of PLOS ONE while writing table title and number

A16: The table has been broken down into two parts and the tables have been labeled according to PLOS ONE guidelines.

Q17. Sources of information is equally important for health seeking behavior

A17: Separate table has been made to highlight the health seeking behavior (Table 3).

Q18: The researchers include Health-seeking Behavior in the title but could not see in any table.

A18: Separate table has been made to highlight the health seeking behavior. Table 3

Q19: Make single sentence

A19: The sentences have been merged.

Reply to comments by Reviewer 2

I. Reply to overall comments

Abstract:

Q1: Suggest using health-seeking behavior, adolescent sexual and reproductive health, teenage pregnancy, influencing factors, and the Chepang population as key words for the paper.

A1: This has been done. Lines 37-38.

Q2: The age group chosen for the study, 15-20 years, seems to differ from commonly used definitions of adolescence (15-19 years), and this warrants clarification. The inclusion of women aged 20 or above in the study should be explained, or alternatively, the data analysis should be revised to maintain consistency with established age categories for adolescents.

A2: Adolescent pregnancy percentage is defined as “Percentage of women aged 15-19 who have given birth or are pregnant with their first child”. As our study measures the prevalence of adolescent pregnancy in the last one year, the operational definition of adolescent pregnancy for this study is as follows:

a. Women aged 15-20 years who have had a live birth within the last 1 year OR

 b. Women aged 15-19 years who are pregnant with their first child during the time of the study 

 The reason our study considers women up to 20 years of age is as follows:

As we are measuring current age of the participants, women who were 19 years of age and delivered last year may be 20 years of age at the time of the study. To include those women, the population group has been mentioned as women aged 15 to 20 years of age. But in essence, the study will still include only adolescents as those women were 19 years old when they delivered.

Q3: While the abstract highlights the role of education as a significant factor in teenage pregnancy among the Chepang community, this point isn't adequately supported in the results and discussion section of the manuscript. It's essential to align the abstract with the study's findings and offer a brief outline of the methodology, including statistical significance.

A3: Our study showed that educational status of the mother and the participant had a statistically significant relationship with adolescent pregnancy. That is why it is highlighted in the abstract. Changes have been made in the manuscript so that it is equally appreciable in the result and discussion. Methodology part has been added.

Introduction:

Q4:The introduction should provide a clearer and more coherent background context for the study. It currently relies on outdated data from NDHS 2016 when more recent data is available (NDHS 2021). If this study predates the NDHS 2021, considering the MICS 2019 results for the problem statement might be more appropriate. It's also suggested to offer a comparative overview of adolescent pregnancy in Nepal, supported by evidence and factors associated with it. Please refer this link https://www.prb.org/wpcontent/uploads/2022/02/Adolescent_Fertility_Atlas_Nepal.pdf for getting more overview of the adolescent fertility.

A4: Changes have been made and data updated using NDHS 2022.

Q5: For the Chepang population, it is essential to verify information with the National Population and Housing Census 2021 report.

A5: The data has been updated according to the recent census report 2021.

Q6: The study title emphasizes health-seeking behavior, but this connection to adolescent pregnancy and relevant influencing factors needs to be more explicitly established.

A6: Information about health seeking behavior and its importance have been added. Lines 75-88.

Materials and Methods:

Q7:The choice of the 15-20 age group for the study should be rationalized, and bias control strategies for age group should be explained. 

A7: Explained in answer to Q2 above.

Q8: The significance level of p=0.17 requires clarification.

A8: This is not the p value but the prevalence for the calculation of sample size. Lines 109-118.

 Q9: The purpose of selection of the study site and the choice of only 3 wards from 7 wards should be justified.

A9: It was not feasible for the researchers to do the study in all the wards due to financial restrictions. Hence we decided to choose three out of seven wards based on our convenience. However, the choice of the specific wards was based on lottery method. 

Q10: To enhance reader-friendliness, provide an overview of variables in table form with operational definitions for both dependent and independent variables.

A10: A table has been added and included in supplementary files (S1 Table).

Q11: Clarify the criteria for defining adolescent pregnancy as “adolescent age of 15-20”, ensuring technical accuracy.

A11: Clarified in response to similar comment above in Q2.

Q12: The sample population details should specify whether only married adolescent girls were included or if unmarried individuals were considered and the reasons for this choice.

A12: Since we wanted to assess the prevalence of adolescent pregnancy the study sample included all the adolescents fitting the eligibility criteria (whether married or unmarried). The questionnaire later assessed their marital status and some of the further analysis were done only in married women.

Results:

Q13: Clarify whether the contraceptive use percentages are among married adolescent girls/women.

A13: Yes, the ‘contraceptive use percentages’ was assessed only among married adolescents.

Q14: Provide the number of adolescent pregnant individuals, which is currently absent in the table.

A14: Apologies for the error. The correction has been made.

Q15: Explain the ANC status among pregnant adolescents and relate it to national protocols on pregnancy and antenatal care indicators. This will help to relate health-seeking behaviour among the adolescent for their care and wellbeing.

A15: This change has been made. 

Q16: When discussing the association of adolescent pregnancy with socio-demographic and family variables, acknowledge the wider confidence intervals and the limitations caused by the small sample size, if relevant. 

A16: We have discussed the repercussions of smaller sample size in our limitation section.

Q17: Specify if the analysis is based solely on married participants.

A17: Since we wanted to assess the prevalence of adolescent pregnancy the study sample included all the adolescents fitting the eligibility criteria (whether married or unmarried). The questionnaire later assessed their marital status and some of the further analysis were done only in married women.

Q18: The examination of health-seeking behavior is currently missing, this should be more detailed.

A18: This component has been added. Lines 211-217.

Discussion:

Q19:The discussion should offer a more in-depth analysis of the factors influencing teenage pregnancy, connecting them to previous studies.

A19: Some additional references have been added/updated.

 Q20: Address statistically insignificant relationships between variables and be honest to explain data, methodology limitations and recommend the need for further research. The sample size is too small to generalize the result.

A20: The sample size was calculated according to norm for a cross sectional study. We have mentioned the limitation of the study and recommendation for further studies. Lines 307-312.

Q21: Teenage pregnancy is often associate with the child marriage in the context like Nepal and child marriage is a result of lack of access to education, information and rights, sexual and reproductive health services etc. which influences individual behaviours. It is important to discuss why child marriage is prevalent there. How has education affected adolescent’s and their family’s health-seeking behaviour? What makes them happy to have a baby as their early age? As health-seeking behaviour is related to individual and community socio-economic and cultural factors. Also, the husband’s education, economic status affects the adolescent pregnancy and health-seeking behaviour which have not been assessed while mother's education and its association with their daughter teenage pregnancy has been briefly touched. I suggest considering all these factors to make a strong argument and scientific explanation for the recommendation. I suggest the author visit global and national report to comprehend the knowledge with the research findings. Include possible alternative explanations and elaborate on policy recommendations, as initially intended.

A21: Thank you for the suggestions. We have added some references to the discussion section in order to make it more rich. However, though the points you have made are very relevant and important in context of teenage pregnancy. I believe exploring some of them is out of scope of this paper. For example, as we have not assessed education and income of the husband in our study, hence we are unable to include this in the discussion. 

Q22: Include possible alternative explanations and elaborate on policy recommendations, as initially intended.

A22: Some changes have been made in the discussion section and recommendation has been added.

References: 

Q23. Ensure that the citation and referencing align with the journal's requirements, preferably using the Vancouver style. Rely on more scientific articles than newspaper to strengthen the arguments and consider thorough English language editing for better comprehension and logical flow.

A23: Necessary changes have been made.

II. Reply to comments in the pdf file

The comments in the pdf file was similar to that in the summary, so the answer to that has been covered above.

Reply to comments by Reviewer 3

I. Reply to overall comments 

Q1: Some of the portion of research methodology need to be clearly explained (inclusion criteria specially) and the result section.

A1: Necessary changes have been made.

II. Reply to comments in the pdf file

Abstract:

Q2: The findings regarding the second objective-health seeking behaviours are not mentioned which need to be considered.

A2: This has been added. Lines 32-33.

Introduction:

Q3. As the recent NDHS 2022 report is readily available, it is recommended to include the most recent ones.

A3: This change has been made. We have revised the introduction using the recent data from the NDHS 2022 and National Health Census 2021. 

Q4: 1. According to the title and the objectives of the study, the sample should be all the pregnant adolescent girls. Then only the factors associated for adolescent pregnancy and their health seeking behaviour could be explored. 

If not, objectives need to be modified. It may be:1.to find out Prevalence of adolescence pregnancy, 2.to explore the factors associated with adolescence pregnancy and 3. their health seeking behaviour among Chepang women' 

A4: In order to know the factors associated with adolescent pregnancy we needed to first find out the prevalence of adolescent pregnancy, hence we have taken adolescent females of reproductive age group for the study as a sample (regardless of pregnancy status).

Here,

Numerator will be either Women aged 15-20 years who have had a live birth within the last 1 year OR Women aged 15-19 years who are pregnant with their first child during the time of the study

Denominator: All adolescent women aged 15 to 20 years. (Reason for taking 20 years is discussed previously)

The objectives are in fact similar to what you have mentioned. For clarity the objectives have been revised.

Q5: According to the objectives, all the adolescent girls need to be included as the sample (sample size calculation is not needed) and then the above set objectives could be explored. Using the prevalence of previous study, if sample size determination was done, there will be sample selection bias. It has to be addressed in this section how it has been controlled or minimised.

A5: Yes, it would be better to include all the adolescents of the study site rather than calculating the sample size. But this was a small study which was funded by the researchers themselves and it was not feasible for us to include all the adolescent girls in our study. Hence the sample size has been calculated. 

Furthermore, the limitation of the study has been explained in lines 307-312.

Methodology

Q6: The inclusion and exclusion criteria need to be explained clearly. Only providing the operational definition of adolescent pregnancy does not suffice. It gave the information that only adolescent pregnancy (those who are pregnant and/or given birth within a year) were included in the study but the result section revealed that irrespective of their pregnant status, all the adolescent girls were included.

A6: The inclusion and external criteria is mentioned in lines 107-109.

Inclusion criteria: All Chepang women aged 15-20 years who gave consent to participate in the study.

Exclusion criteria: The study participants who were unable to answer due to language barrier 

The sentence has been revised for better understanding.

For your kind consideration, we have included all the adolescents and not pregnant adolescents.

Results

Q7: It is suggested to use a uniform word to address the sample of the study. Either respondent or participant.

A7: The change has been made to maintain uniformity. The word ‘respondent’ have been replaced by ‘participant’.

Q8: It would be better to present the data in a way that among 217 participants how many of them were the targeted participants (who meet the inclusion criteria of adolescent pregnancy) either in words or in table.

A8: All the 217 participants were adolescents in the age group 15 to 20 years of age. All of them met the inclusion criteria. 

Q9: The sum is 99.8 not 100

A9: The change has been made. The total is now 100.

 Q10: Total is 99.9

A10: The change has been made. The total is now 100.

Q11: Total is 100.1

A11: The change has been made. The total is now 100.

Q12: Mention whether it is per week, month or per annum.

A12: This is monthly income. The change has been made.

Q13: Is it related to health seeking behaviour then separate title can be given as it is one of the objectives? 

A13: This has been done.

Q14: It would be better to operationally define what is health seeking behaviour? Because the highlighted sentence show only hospital related factors as other non-hospital related factors also could have been followed or done by them.

A14: Definition of Reproductive health seeking behavior has been operationalized. Lines 137-140. 

Q15: Mention either % or percent.

A15: This change has been made.

Q16: It is suggested to mention adolescent pregnancy even in result section as they are one of the main or targeted samples among whom the objectives of the study are based on.

A16: This has been mentioned in the result section. Lines 204-209.

Q17: Write in sentence format.

A17: This change has been made. 

Q18: This section need to include answers to the set objectives in a simpler sentence even a lay person can understand it. The answers to other objective of health seeking behaviour seems missing.

A18: This change has been made.

---

## [Editor Report · Decision Letter 1]

11 Jan 2024

PONE-D-23-26050R1Factors associated with adolescent pregnancy among Chepang women and their health-seeking behavior in Ichchhakamana rural municipality of Chitwan districtPLOS ONE

Dear Dr. Pant,

Thank you for submitting your manuscript to PLOS ONE. After careful consideration, we feel that it has merit but does not fully meet PLOS ONE’s publication criteria as it currently stands. Therefore, we invite you to submit a revised version of the manuscript that addresses the points raised during the review processPlease ensure that your decision is justified on PLOS ONE’s publication criteria and not, for example, on novelty or perceived impact.

We look forward to receiving your revised manuscript.

Kind regards,

Umesh Raj Aryal, PhD

Academic Editor

PLOS ONE

Journal Requirements:

Additional Editor Comments:

Correct Spelling in "Fisher Exact Test". Next , VIF values should be presented in result section. Why author did not consider non-response rate in sample size?

---

## [Author Response · Author response to Decision Letter 1]

22 Jan 2024

Journal Requirements:

Comment1: Please review your reference list to ensure that it is complete and correct. If you have cited papers that have been retracted, please include the rationale for doing so in the manuscript text, or remove these references and replace them with relevant current references. Any changes to the reference list should be mentioned in the rebuttal letter that accompanies your revised manuscript. If you need to cite a retracted article, indicate the article’s retracted status in the References list and also include a citation and full reference for the retraction notice.

Answer: I have revised the references to correct some errors in links and also ensured that the references are correct. I have also removed all the hyperlinks and edited them according to journal guidelines. Additionally, I also checked if any of the cited references were retracted. But there were no retracted manuscripts.

Additional Editor Comments:

Comment 2: Correct Spelling in "Fisher Exact Test". 

Answer: The correction has been made. (Lines 26, 153,242)

Comment 3: VIF values should be presented in result section. 

Answer: The table of Collinearity Statistics has been added as a table in supplementary files (S2 table). 

Comment 4: Why author did not consider non-response rate in sample size?

Answer: I apologize for the error. This portion was omitted by mistake in the last revision.

The calculated sample size was 216.8

Considering 10% non- response rate, total sample size = 216.8/0.9 = 240.8≈ 241

Therefore, the total sample size was 241.

However, Out of the 241 women chosen, a total of 217 participants responded, yielding a response rate of 90%. .

According to this information, necessary change has been made in the methodology (lines 120-121, 127-128) and result section of the manuscript (Lines 180).

---

## [Editor Report · Decision Letter 2]

26 Feb 2024

PONE-D-23-26050R2Factors associated with adolescent pregnancy among Chepang women and their health-seeking behavior in Ichchhakamana rural municipality of Chitwan districtPLOS ONE

Dear Dr. Pant,

Thank you for submitting your manuscript to PLOS ONE. After careful consideration, we feel that it has merit but does not fully meet PLOS ONE’s publication criteria as it currently stands. Therefore, we invite you to submit a revised version of the manuscript that addresses the points raised during the review process.

**ACADEMIC EDITOR: Please insert comments here and delete this placeholder text when finished.**Website link of reference no 17 and 27 needs to be revised.

We look forward to receiving your revised manuscript.

Kind regards,

Umesh Raj Aryal, PhD

Academic Editor

PLOS ONE

Journal Requirements:

Additional Editor Comments (if provided):

Web site link for reference no 17  and 27 needs to revised.

---

## [Author Response · Author response to Decision Letter 2]

9 Mar 2024

Response to editor:

Comment 1: Website link of reference no 17 and 27 needs to be revised.

Answer 1: Thank you for the suggestion. The correction has been made. (lines 399 and 433-434)

---

## [Editor Report · Decision Letter 3]

14 Mar 2024

Factors associated with adolescent pregnancy among Chepang women and their health-seeking behavior in Ichchhakamana rural municipality of Chitwan district

PONE-D-23-26050R3

Dear Dr. Pant,

We’re pleased to inform you that your manuscript has been judged scientifically suitable for publication and will be formally accepted for publication once it meets all outstanding technical requirements.

Kind regards,

Umesh Raj Aryal, PhD

Academic Editor

PLOS ONE

Additional Editor Comments (optional):

No

Reviewers' comments:

No

---

## [Editor Report · Acceptance letter]

19 Mar 2024

PONE-D-23-26050R3 

PLOS ONE

Dear Dr. Pant, 

I'm pleased to inform you that your manuscript has been deemed suitable for publication in PLOS ONE. Congratulations! Your manuscript is now being handed over to our production team.

Kind regards, 

on behalf of

Dr. Umesh Raj Aryal 

Academic Editor

PLOS ONE